# Self-reported medication adherence among patients with diabetes or hypertension, Médecins Sans Frontières Shatila refugee camp, Beirut, Lebanon: A mixed-methods study

**Mariam Mohamad**[1]*, **Krystel Moussally**[2,3☺], **Chantal Lakis**[4☺], **Maya El-Hajj**[5], **Sola Bahous**[6], **Carla Peruzzo**[4], **Anthony Reid**[7], **Jeffrey K. Edwards**[8]

**1** Field mission, Médecins Sans Frontières, Operational Center Brussels, Shatila, Beirut, Lebanon, **2** Lebanon branch office, Médecins Sans Frontières, Beirut, Lebanon, **3** Middle-East Medical Unit (MEMU), Médecins Sans Frontières, Beirut, Lebanon, **4** Coordination, Médecins Sans Frontières, Operational Center Brussels, Beirut, Lebanon, **5** Clinical and Epidemiological Research Laboratory, Faculty of Pharmacy, Lebanese University, Hadath, Lebanon, **6** School of Medicine, Lebanese American University, Beirut, Lebanon, **7** Operational Research Unit, Médecins Sans Frontières, Operational Center Brussels, Luxembourg City, Luxembourg, **8** Department of Global Health, University of Washington, Seattle, Washington, United States of America

☺ These authors contributed equally to this work.
* mariammohamad02@gmail.com

**Data Availability Statement:** Data sets used to generate the results of the current study were not

## Abstract

### Introduction

Low adherence to medications, specifically in patients with Diabetes (DM) and Hypertension (HTN), and more so in refugee settings, remains a major challenge to achieving optimum clinical control in these patients. We aimed at determining the self-reported medication adherence prevalence and its predictors and exploring reasons for low adherence among these patients.

### Methods

A mixed-methods study was conducted at Médecins Sans Frontières non-communicable diseases primary care center in the Shatila refugee camp in Beirut, Lebanon in October 2018. Data were collected using the validated Arabic version of the 8-items Morisky Medication Adherence Scale (MMAS-8) concurrently followed by in-depth interviews to explore barriers to adherence in patients with DM and/or HTN. Predictors of adherence were separately assessed using logistic regression with SPSS© version 20. Manual thematic content analysis was used to analyze the qualitative data.

### Results

Of the 361 patients included completing the MMAS, 70% (n = 251) were moderately to highly adherent (MMAS-8 score = 6 to 8), while 30% (n = 110) were low-adherent (MMAS-8

made publicly available due to the sensitivity of the context we work in – Syrian refugees that might not be documented and illegally present in the country. In fact, ethical risks include, but are not limited to, the nature of Medecins Sans Frontieres (MSF) operations and target populations being such that data collected are often highly sensitive. Due to ethical and legal restrictions imposed by MSF operational authorities, MSF is obliged towards its patients to protect their data responsibility. Data will be available on request in accordance with MSF's data sharing policy (available at: http://fieldresearch.msf.org/msf/handle/10144/306501). Requests for access to data should be made to data.sharing@msf.org.

**Funding:** The author(s) received no specific funding for this work.

**Competing interests:** The authors have declared that no competing interests exist.

score<6). Patients with DM-1 were the most likely to be moderately to highly adherent (85%; n = 29). Logistic regression analysis showed that patients with a lower HbA1C were 75% more likely to be moderately to highly adherent [(OR = 0.75 (95%CI 0.63–0.89), p-value 0.001]. Factors influencing self-reported moderate and high adherence were related to the burden of the disease and its treatment, specifically insulin, the self-perception of the disease outcomes and the level of patient's knowledge about the disease and other factors like supportive family and healthcare team.

## Conclusion

Adherence to DM and HTN was good, likely due to a patient-centered approach along with educational interventions. Future studies identifying additional factors and means addressing the barriers to adherence specific to the refugee population are needed to allow reaching optimal levels of adherence and design well-informed intervention programs.

## Introduction

Non communicable diseases (NCDs), such as diabetes mellitus (DM) and hypertension (HTN) are an increasing burden globally, especially in the Middle-East region [1]. In Lebanon, where an estimated 1.5 million Syrians have been displaced following the onset of the Syrian war in 2011, another estimated 500,000 Palestinians are settled and more than 90% of people lose their lives secondary to an NCD [2]. Médecins Sans Frontières (MSF) provides free primary care for refugees in the Shatila camp in Southern Beirut using a multidisciplinary and comprehensive model of care extensively described elsewhere [3]. In the MSF Shatila clinic, as of September 2019, 84% of patients attending the clinic have DM (types 1 or 2) and/or HTN. While a recent study, different in its scope and results, conducted at the MSF Shatila clinic showed promising outcomes of care in DM and HTN patients [3], it has been challenging for MSF providers to achieve optimum control in these patients.

One of the determinants of poor NCD outcomes is low adherence to medication, which has been estimated globally to be 50% in patients with DM and/or HTN, making it a primary risk factor for poor outcomes in these diseases [4]. Although low adherence to medication is a global problem and a modifiable behavioral factor that can lead to decreased morbidity and mortality, it is still repeatedly overlooked, when it may have more impact on clinical outcomes than any other treatment advances, especially in low-resource settings [4–6]. Additionally, some factors that might affect medication adherence are potentially specific to a refugee setting compared to other contexts [7]. In fact, while adherence to medication is multifactorial, cultural beliefs, access to healthcare, individual characteristics and patient-related factors are all known factors affecting it [4, 8]. Therefore, there is a desperate need for context-specific evidence related to low adherence to medications, more so in the refugee population of the Middle-East, which carries a high burden of NCDs, and on which literature is scarce.

To our knowledge, only one study has looked into medication adherence rates and the associated risk factors among patients with NCDs in a refugee setting—in Palestinians living in Jordan [9]. However, this study was observational in nature and did not explore the challenges faced from patients' perspective. Another study recently published mentioned that around 25% of Syrian refugees in Lebanon had to interrupt their HTN or DM medication in the year preceding the survey due to costs [10]. In addition, a study evaluating MSF primary care for

NCDs in Jordan showed a high level of self-reported adherence to NCDs in the Syrian refugee and Jordanian population. It recommended exploring these results as patients and health staff have reported intentional and unintentional low adherence behaviors contradicting the rate of self-reported adherence [11]. This makes it crucial to understand the factors related to low adherence to medications in the Shatila refugee setting to allow appropriate adaptation of the model of care and ultimately better clinical outcomes for those being served [12].

The aim of this study was to determine: 1. self-reported medication adherence rates, 2. risk factors for low adherence, and 3. reasons for low adherence among patients with DM and/or HTN seen at the MSF Shatila primary care center in Beirut, Lebanon.

## Methods

### Study design

This study was a mixed-methods design carried out between the 1st and 19th of October 2018 in MSF NCD Shatila clinic. It included a cross-sectional quantitative component with qualitative in-depth interviews.

### Cross-sectional component

The 8-item Morisky Medication Adherence Scale (MMAS-8) questionnaire was used to determine the prevalence of self-reported adherence among patients with DM and/or HTN presenting at the Shatila clinic during the study period [13]. The first seven questions have a yes/no answer, while the last question uses a 5-point Likert scale. Each answer is scored 0 or 1. The MMAS-8 has been previously used for a wide range of diseases and is validated for both DM and HTN [9, 14, 15]. More recently, the MMAS-8 was used in studies assessing self-reported adherence in Arabic speaking populations, such as the United Arab Emirates (UAE), Saudi Arabia, Jordan, and Palestine [5, 9, 16–18]. The Arabic version of the MMAS-8 was tested for its reliability and validity for type 2 DM patients in Libya [19]. In addition to the MMAS-8, socio-demographic and clinical variables were collected and used to determine the predictors of non-adherence to medications among the study patients. Socio-demographic variables consisted of: age, gender, area of residence (in or outside the program catchment area), nationality, refugee registration status (yes/no), total years of displacement from home country ($\leq$ 12 months, >12–$\leq$ 24 months, >24 months), employment (yes/no), and literacy (illiterate, reads and/or writes). Clinical variables included: diagnosis (type 1 DM only, type 2 DM only, HTN only, combined DM and HTN), time since diagnosis, patient support and education counseling (yes, with number of sessions, or no), number of prescribed chronic medications (limited to those prescribed by MSF), most recent Hemoglobin A1C (HbA1C) measurement, insulin use (yes/no) for DM patients, and blood pressure recording at the time of the appointment for patients with HTN.

### Qualitative component

In-depth interviews were conducted on a subsample of participants who had completed the MMAS-8, based on their adherence score, using a pre-drafted interview questionnaire exploring the barriers and facilitators of adherence. The questionnaire used was piloted prior to the actual study data collection.

### Setting

Shatila is a Palestinian refugee camp in south Beirut that was initially created in 1949 to accommodate refugees fleeing from the northern parts of Palestine. It comprises an area of

approximately one-square kilometer and now supports refugees from Syria as well. Estimates of the population living in Shatila are uncertain, but it is estimated to be more than 25,000 people in this high-density area, of whom approximately 18,000 are Syrians. Shatila is characterized by an extreme rate of unemployment, insecurity, and limited infrastructure making living conditions challenging and reliance on private donors and non-governmental organizations (NGOs) substantial, especially in the absence of any type of supporting healthcare services within the camp.

## MSF model of care

MSF initiated the primary health care center in Shatila in September 2013 providing, among other services, NCD services free of charge to vulnerable refugees and the host community for patients with DM, HTN, cardiovascular disease, asthma, epilepsy and/or hypothyroidism.

Since care provided in MSF center is free of charge for the above-mentioned diseases, we believe that the patients followed-up for care at our center do not seek care elsewhere. However, this cannot be verified. The model of care used for DM and HTN patients consists of five components: case management, patient support and education counseling (PSEC), health promotion, mental health care and social work. Doctors are in charge for consultation provided for all the patients, there is no hired specialized doctor so in case a patient needs such service he/she will get guidance for one outside MSF clinic. Nurses are in charge of checking patients' vital signs such as blood glucose and blood pressure and of doing some needed lab tests. Patients followed at our center and who are stable as per their blood pressure -for patients with HTN- and/or their blood sugar readings -for those with DM-, have their medical consultation and medication refill taking place every 2 to 3 months. Those who are unstable, are requested to present for an appointment within a month in order to closely follow them up and monitor their outcomes and treatment plans. In addition, mental health sessions are provided by a psychologist and sessions are available for who are in need even if they are not MSF patients. Social work's main task is to refer or guide MSF patients to other NGOs or associations that provide services not available in MSF. Finally, health promotion sessions are available, both individual and in groups, on health-related topics. A full description of this model can be found elsewhere [3].

The PSEC component aims at a more individualized education and counseling approach for patients with uncontrolled DM, those newly diagnosed with DM, and for those who need insulin as part of their treatment plan. PSEC is provided only for DM patients due to limited program capacity and resource prioritization. Patients are referred to PSEC by a doctor based on the above-mentioned criteria and clinical judgment. PSEC follows a standardized and contextually adapted educational package for diabetes that includes education, support and counseling on the disease process and its complications, medication adherence, self-monitoring of blood glucose and lifestyle habits with diet instructions. It is a patient-centered package that is customized to the resources available for refugees and allows, through the provision of individual sessions, a space for patients to discuss challenges related to their treatment or disease.

Individual objectives are set at each PSEC session for every patient. Up to seven sessions are provided in order to complete the educational package. The PSEC team involves, whenever possible, family members or caregivers in order to create an enhanced atmosphere of support. Patients are discharged from the PSEC when they have completed all necessary modules, independent of their HbA1C. At the time of the study, less than 50% of the patients with DM alone or with DM and HTN had been referred to PSEC.

As part of the case management of patients with DM, patients in need of insulin are provided with syringes and necessary supplies. Insulin pen devices started to be provided in September 2018 for type 1 DM patients ≤ 15 years old. Both rapid and slow-acting insulin were provided, depending on the patient's requirements.

## Study population and sample size

Eligible patients for the cross-sectional component were those who, at the time of the study: 1) were diagnosed with type 1 or 2 DM and/or HTN and on follow-up for their disease in the MSF NCD Shatila clinic; 2) were on ≥1 chronic medication for their DM and/or HTN; and 3) gave their written consent to take part in the study. Assuming a conservative non-adherence rate of 50%, a margin error of 5%, and a 95% confidence interval, a total of 343 patients was needed to achieve 80% power. The sample size was inflated by 10% for an estimated non-response rate and an additional 10% to account for no-show at the time of the scheduled appointment, thereby increasing the sample size to 412 patients.

A random sampling stratified by type of disease and proportionally allocated to the four disease subgroups (type1 DM, type 2 DM, HTN, and DM with HTN) was applied in order to ensure an adequate representation of the target diseases at the NCD clinic and to improve precision of the overall estimate of self-reported low adherence. The random list of patients was computed based on the inclusion criteria from the clinic NCD list of patients who were expected for their next appointment during the time of the data collection. The list was shared with the NCD doctors who directed selected patients to the PSEC counselor responsible for obtaining the consent and performing the MMAS-8 questionnaire and conducting the interviews on the qualitative sample. For the patients who were <15 years old, the MMAS-8 questionnaire, and the interview were carried out with their caregiver (a parent or a relative living with the patient who knows their medical history and takes care of their medication intake). For patients or caregivers who were unable to read and/or write, the MMAS-8 questionnaire was verbally administered. The verbal administration of the MMAS-8 questionnaire has been previously used in other studies in communities with lack of or low-literacy level [20, 21].

Out of the sample of patients who self-completed the MMAS questionnaire, a purposive sub-sample of 30 patients was selected for in-depth interviews, in order to ensure a representative mix of participants with regard to gender, disease type, age, and adherence status as per their MMAS-8 score. Interviews were conducted by the PSEC counselor in Arabic, the native language of the patients; a standardized interview questionnaire was used as a guide was audio-taped with the written consent of the patients. Thirty interviews were carried out and saturation was reached. Interviews took up to 30 minutes per participant. The PSEC counselor was a woman trained in conducting qualitative interviews and known to the patients.

## Statistical analysis

Quantitative and qualitative analyses were conducted separately. For the cross-sectional data analysis, adherence to medications was considered as the dependent variable. High adherence was defined with a MMAS-8 score of 8, while moderate and low adherence were defined with a MMAS-8 score of 6 to 7 and <6, respectively. Although there are no published guidelines or recommendations for the MMAS-8 questionnaire in the literature defining categorization of adherence, these cut-offs were chosen based on a similar representation in other studies [22, 23]. Comparison of patients with a low level of adherence versus moderate or high level of adherence was done using bivariate analysis, considering that a moderate level of adherence is clinically acceptable as they are more likely to be in control of care than those who are less adherent [24]. Also, it is believed that this level is more feasible as a target knowing that the

population we serve face contextual challenges in the access to care, medications, and follow-up of their treatment. Diagnostic and clinical data were collected directly from the patients' files at the MSF clinic while data regarding demographic or adherence were collected by a semi-structured questionnaire directly from the patients. Employment data was collected on patients who were ≥ 15 years old. Patients were considered literate if they were capable of reading and/or writing. As per MSF clinical guidelines, controlled DM was defined as having an HbA1C value of < 8%, and controlled HTN as having a blood pressure < 140/90mmHg. The choice of an HbA1C target of < 8% compared to the international standard of 7% has been explained elsewhere [3].

Descriptive statistics were used to report the characteristics of the study population. Demographic and clinical characteristics of patients with different levels of adherence were compared using appropriate statistical testing. Comparison of patients with a low level of adherence versus those with moderate or high level of adherence was done using bivariate analysis. Predictors of adherence for patients with DM and those with HTN were separately assessed using logistical regression with a cut-off of 6 for the MMAS-8. The model was adjusted for variables with a p-value <0.2 on the bivariate analysis. Analysis of the quantitative data was done using SPSS© version 20. P-values < 0.05 were considered significant.

Qualitative data were transcribed from the audio recordings and translated verbatim then analyzed manually using a thematic approach. Analyses were done by the first three authors in the study to ensure alignment in data interpretation. They conducted a thematic analysis of all interviews transcripts and coded each answer in order to highlight patients' statements and their point of views about each item discussed in the questionnaire and provide a possible explanation for the different perspectives. The findings were reported by using Consolidated Criteria for Reporting Qualitative Research format [25]. The results of the quantitative and the qualitative analysis were triangulated at the interpretation stage.

### Ethics

Ethics approval was obtained from the MSF Ethics Review Board (Geneva, Switzerland) and the Lebanese American University Institutional Review Board.

## Results

### Characteristics of the study population

Of 412 patients randomly selected for the quantitative component, 361 were included in the study (Fig 1). The excluded 51 patients were those who did not show up at their scheduled appointment (n = 43, 84%) or refused to participate (n = 8, 16%).

At intake, the median age of the patients included was 54 years (Interquartile range [IQR]: 45–61); the majority were Syrian refugees (n = 330, 91%), and were female (n = 214, 59%). Patients with both DM and HTN were the most prevalent (n = 161, 45%). At the time of the study, 38% (n = 107) of DM patients used insulin, and almost half had at least one PSEC session (n = 137, 48%). Most were unemployed, more than 50% had more than one chronic disease and most were taking more than four chronic medications. All the characteristics of the participants are described in Table 1.

### Adherence to medication—Quantitative results

Based on the MMAS-8 questionnaire results the median score for all participants was of 7 (IQR 5, 8). Of the participants, 31% (n = 111) were highly adherent with a score = 8 at the time of the study, while 39% (n = 140) had a score of 6 to 7 and 30% (n = 110) had a score of <6.

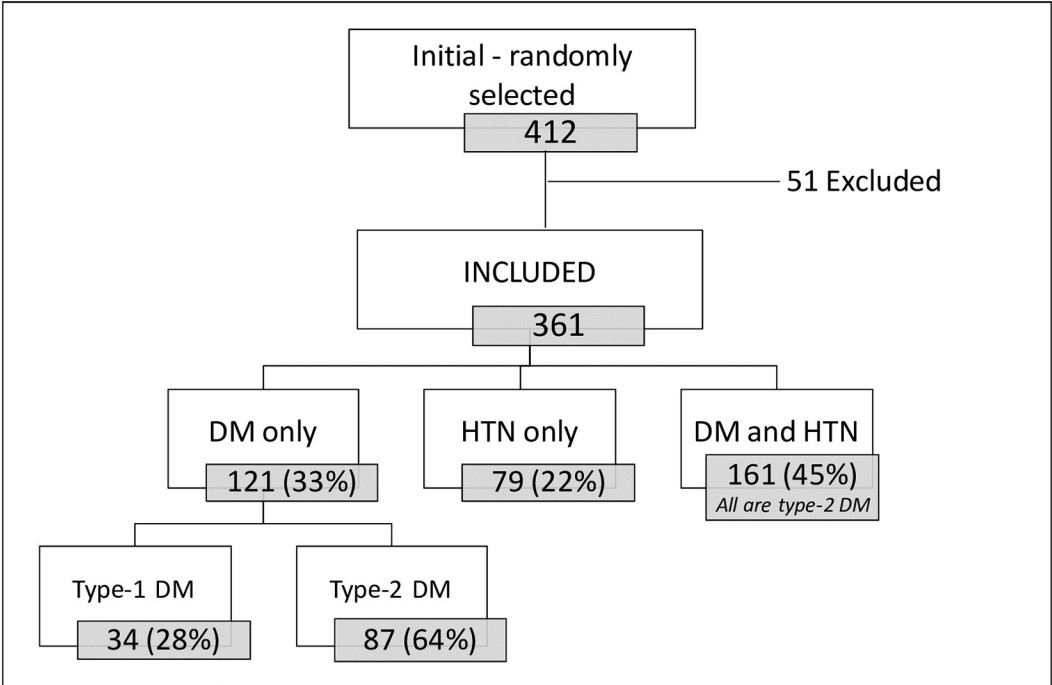

**Fig 1. Flowchart of study participants among patients with diabetes and/or hypertension, Médecins Sans Frontières Shatila refugee camp, Beirut, Lebanon, October 2018.**

Out of the patients using insulin at the time of the study, 13% (n = 14/107) had high adherence while the majority (55%, n = 59/107) were moderately adherent. For patients with DM, their HbA1C mean decreased with an increased level of adherence to reach 7.5% for patients who were highly adherent (Table 1). Out of the patients with HTN, 34% (n = 82/240) had a high adherence level. Level of adherence seems to increase with a decreased time since diagnosis (Table 1).

The most prevalent contributor was an intentional cut back of medicine by the patient when feeling worse/better when taking the medicine [Yes on questions 3 and/or 6; (n = 138, 38%)], followed by an un-intentional interruption by 25% (n = 91) of patients who stated that they sometimes forget to take their medicine (Table 2). Out of those who intentionally cut back their medication, 107 (88%) were patients with DM and among those patients with DM, the majority (n = 66, 62%) were insulin-dependent.

## Bivariate analysis

Results of the bivariate analysis showed that patients who were >15 years old tended to be less moderate to highly adherent compared to those who are 15 years old or younger (p-value = 0.018) (Table 3). In relation to diagnosis type, patients with DM-1 were the most likely to be moderately to highly adherent (85%) and those with HTN or DM and HTN the least likely (63% and 65%, p-value 0.027) (Table 3). Being on two chronic medications seemed to be significantly increasing the likelihood of being moderately to highly adherent. (Table 3). Adherence was also significantly higher among those who had controlled DM (HbA1C<8%). HTN control was not significantly related to adherence for HTN patients (Table 3).

**Table 1. Description of the general characteristics of the participants, stratified by their level of adherence, Shatila primary care clinic, Beirut, October 2018.**

| Characteristics at time of study | Total | Low adherence MMAS < 6 | Moderate adherence MMAS = 6–7 | High adherence MMAS = 8 |
|---|---|---|---|---|
| | (N = 361) | (n = 110) | (n = 140) | (n = 111) |
| **Age—year—median, (IQR)** | 54 (45–61) | 53 (46, 63) | 53 (41, 60) | 54 (48, 62) |
| **Age categories—year—n (%)** | | | | |
| ≤15 | 18 (5) | 1 (6) | 16 (88) | 1 (6) |
| >15 | 343 (95) | 109 (32) | 124 (36) | 110 (32) |
| **Gender—n (%)** | | | | |
| Male | 147 (41) | 37 (25) | 63 (43) | 47 (32) |
| Female | 214 (59) | 73 (34) | 77 (36) | 64 (30) |
| **Nationality—n (%)** | | | | |
| Syrian | 330 (91) | 103 (31) | 124 (38) | 103 (31) |
| Lebanese | 21 (6) | 5 (24) | 9 (43) | 7 (33) |
| Palestinian | 10 (3) | 2 (20) | 7 (70) | 1 (10) |
| **Employment[a]—n (%)** | | | | |
| Yes | 62 (23) | 14 (23) | 30 (48) | 18 (29) |
| No | 210 (77) | 75 (36) | 68 (32) | 67 (32) |
| **Literacy—n (%)** | | | | |
| Illiterate | 154 (43) | 53 (34) | 55 (36) | 46 (30) |
| With some education | 207 (57) | 57 (28) | 85 (41) | 65 (31) |
| **Diagnosis—n (%)** | | | | |
| DM-1 Only | 34 (9) | 5 (15) | 27 (79) | 2 (6) |
| DM-2 Only | 87 (24) | 20 (23) | 40 (46) | 27 (31) |
| HTN Only | 79 (22) | 29 (37) | 21 (27) | 29 (37) |
| DM + HTN | 161 (45) | 56 (35) | 52 (32) | 53 (33) |
| **Chronic morbidities[b]—n (%)** | | | | |
| 1 | 159 (44) | 43 (27) | 70 (44) | 46 (29) |
| > 1 | 202 (56) | 67 (33) | 70 (35) | 65 (32) |
| **Prescribed chronic medications number[c]—n (%)** | | | | |
| 1 | 14 (4) | 5 (36) | 5 (36) | 4 (29) |
| 2 | 65 (18) | 10 (15) | 39 (60) | 16 (25) |
| 3 | 55 (15) | 20 (36) | 17 (31) | 18 (33) |
| ≥4 | 227 (63) | 75 (33) | 79 (35) | 73 (32) |
| **Attended PSEC[d]—n (%)** | | | | |
| Yes | 137 (38) | 42 (31) | 63 (46) | 32 (23) |
| No | 145 (62) | 39 (27) | 56 (39) | 50 (35) |
| **PSEC sessions number[d]—median (IQR)** | 5 (3–6) | 4 (2–5) | 5 (4–6) | 5 (2–5) |
| **Insulin use[d]—n (%)** | | | | |
| No | 174 (62) | 46 (26) | 60 (35) | 68 (39) |
| Yes | 107 (38) | 34 (32) | 59 (55) | 14 (13) |
| **HbA1C[d]—% [mean (SD)]** | 8.2 (1.6) | 8.6 (1.6) | 8.4 (1.7) | 7.5 (1.2) |
| **Blood pressure[e]—mmHg [mean (SD)]** | | | | |
| Systolic | 130 (18) | 131 (22) | 133 (17) | 127 (15) |
| Diastolic | 79 (11) | 78 (12) | 80 (11) | 79 (8) |
| **Controlled blood pressure[e]—n (%)** | | | | |
| Yes | 132 (55) | 48 (36) | 37 (28) | 47 (36) |
| No | 108 (45) | 37 (34) | 36 (33) | 35 (33) |
| **Characteristics at first visit** | | | | |
| **Place of residency—n (%)** | | | | |

(*Continued*)

**Table 1.** (Continued)

| Characteristics at time of study | Total | Low adherence MMAS < 6 | Moderate adherence MMAS = 6–7 | High adherence MMAS = 8 |
|---|---|---|---|---|
| | (N = 361) | (n = 110) | (n = 140) | (n = 111) |
| In catchment area | 101 (28) | 34 (34) | 38 (37) | 29 (29) |
| Outside catchment area | 260 (72) | 76 (29) | 102 (39) | 82 (32) |
| **Time since diagnosis**—year—*median (IQR)* | 7 (3–12) | 8.5 (4–14) | 7 (3–14) | 6 (3–10) |
| **Duration of displacement**—year—*median, (IQR)* | 5 (3–6) | 5 (3–6) | 5 (3–6) | 5 (3–6) |

DM-1 type-1 diabetes, DM-2 type-2 diabetes, HTN hypertension, IQR interquartile range, PSEC patient support education and counseling, SD standard deviation, UNHCR united nations high commission for refugees.

[a] For those between 15 years and 64 years.

[b] Morbidities included, besides diabetes and hypertension: asthma, epilepsy, chronic obstructive pulmonary disease, hypothyroidism and cardiovascular diseases.

[c] Medications related to chronic morbidities recorded.

[d] Calculated for patients with diabetes; most recent HbA1C recorded, on average 2.2 months from the study date

[e] Calculated for patients with hypertension.

## Multivariate analysis

Logistic regression analysis for patients with DM showed that the only statistically significant predictor of moderate/high adherence was the last recorded HbA1C level with patients with a lower HbA1C being 75% more likely to be moderately to highly adherent [(OR = 0.75 (95%CI 0.63–0.89), p-value 0.001]. For patients with HTN, when confounders were accounted for, none of the predictors remained statistically significant.

**Table 2. Proportion of yes/no responses to the MMAS-8 questions by the study participants, Shatila primary care clinic, Beirut, October 2018.**

| MMAS-8 specific question | Yes | No |
|---|---|---|
| | n (%) | n (%) |
| **Question 1**–Do you sometimes forget to take your medicine? | 91 (25) | 270 (75) |
| **Question 2**–Over the past 2 weeks, were there any days when you did not take your medicine? | 109 (31) | 252 (69) |
| **Question 3**–Have you ever cut back or stopped taking your medicine without telling your doctor because you felt worse when you took it? | 122 (34) | 239 (66) |
| **Question 4**–When you travel or leave home, do you sometimes forget to bring along your medicine? | 44 (12) | 317 (88) |
| **Question 5**–Did you take all your medicines yesterday? | 293 (81) | 68 (19) |
| **Question 6**–When you feel like your symptoms are under control, do you sometimes stop taking your medicines? | 37 (10) | 324 (90) |
| **Question 7**–Do you ever feel hassled about sticking to your treatment plan? | 83 (23) | 278 (77) |
| **Question 8**–How often do you have difficulty remembering to take all your medicine? | | |
| Never | 261 (72) | |
| Occasionally | 67 (19) | |
| Sometimes | 15 (4) | |
| Usually | 14 (4) | |
| All the time | 4 (1) | |

**Table 3. Bivariate analyses among low adherence (MMAS-8 score <6) and moderately/high adherence (MMAS-8 score≥6) for study participants, Shatila primary care clinic, Beirut, October 2018.**

| Characteristics at the time of study | Low adherence (n = 110) | Moderate/High adherence (n = 251) | p-value[a] |
|---|---|---|---|
| **Age**—year—*median, (IQR)* | 53 (46–63) | 54 (45–61) | 0.803 |
| **Age categories**—year—*n (%)* | | | |
| ≤15 | 1 (6) | 17 (94) | |
| >15 | 109 (32) | 234 (68) | 0.018 |
| **Gender**—*n (%)* | | | |
| Male | 37 (25) | 110 (75) | |
| Female | 73 (34) | 141 (66) | 0.070 |
| **Nationality**—*n (%)* | | | |
| Syrian | 103 (31) | 227 (69) | |
| Lebanese | 5 (24) | 16 (76) | |
| Palestinian | 2 290) | 8 (80) | 0.657 |
| **Employment**[b]—*n (%)* | | | |
| Yes | 14 (23) | 48 (77) | |
| No | 75 (36) | 135 (64) | 0.053 |
| **Literacy**—*n (%)* | | | |
| Illiterate | 53 (34) | 101 (66) | |
| With some education | 57 (27) | 150 (73) | 0.160 |
| **Diagnosis**—*n (%)* | | | |
| DM-1 Only | 5 (15) | 29 (85) | |
| DM-2 Only | 20 (23) | 67 (77) | |
| HTN Only | 29 (37) | 50 (63) | |
| DM + HTN | 56 (35) | 105 (65) | 0.027 |
| **Chronic morbidities**[c]—*n (%)* | | | |
| 1 | 43 (27) | 116 (73) | |
| > 1 | 67 (33) | 135 (67) | 0.209 |
| **Prescribed chronic medications number**[d]—n (%) | | | |
| 1 | 5 (36) | 9 (64) | |
| 2 | 10 (15) | 55 (85) | |
| 3 | 20 (36) | 35 (64) | |
| ≥4 | 75 (33) | 152 (67) | 0.022 |
| **Attended PSEC**[e]—*n (%)* | | | |
| Yes | 42 (31) | 95 (69) | |
| No | 39 (27) | 106 (73) | 0.485 |
| **PSEC sessions number**[e]—*median (IQR)* | 4 (2–5) | 5 (3–6) | 0.100 |
| **Insulin use**[e]—*n (%)* | | | |
| No | 46 (26) | 128 (74) | |
| Yes | 34 (32) | 73 (68) | 0.336 |
| **HbA1C**[e]—% *[mean (SD)]* | 8.6 (1.6) | 8.0 (1.6) | <0.002 |
| **HbA1C**[e]-% *[n (%)]* | | | |
| <8 | 27 (19) | 116 (81) | |
| ≥8 | 53 (39) | 84 (61) | <0.0001 |
| **Blood pressure**[f]–mmHg, *mean (SD)* | | | |
| Systolic | 131 (21.6) | 130 (16.3) | 0.555 |
| Diastolic | 78 (12.5) | 80 (9.5) | 0.400 |
| **Controlled blood pressure**[f]—*n (%)* | | | |
| Yes | 48 (36) | 84 (64) | |

*(Continued)*

**Table 3.** (Continued)

| Characteristics at the time of study | Low adherence (n = 110) | Moderate/High adherence (n = 251) | p-value[a] |
|---|---|---|---|
| No | 37 (34) | 71 (66) | 0.735 |
| **Characteristics at first visit** | | | |
| **Place of residency**—n (%) | | | |
| In catchment area | 34 (34) | 67 (66) | |
| Outside catchment area | 76 (29) | 184 (71) | 0.411 |
| **Time since diagnosis**—year—*median, (IQR)* | 8 (4–14) | 7 (3–12) | 0.103 |
| **Duration of displacement**—year—*median, (IQR)* | 5 (3–6) | 5 (3–6) | 0.931 |

DM-1 type1 diabetes, DM-2 type 2 diabetes, HTN hypertension, IQR interquartile range, PSEC Patient Support Education and Counseling, SD standard deviation, UNHCR United Nations High Commission for Refugees.

[a] p-value <0.05 is statistically significant.

[b] For those between 15 years and 64 years.

[c] Morbidities included, besides diabetes and hypertension: asthma, epilepsy, chronic obstructive pulmonary disease, hypothyroidism and cardiovascular diseases.

[d] Medications related to chronic morbidities recorded.

[e] Calculated for patients with diabetes; most recent HbA1C recorded, on average 2.2 months from the study date

[f] Calculated for patients with hypertension.

## Adherence–qualitative results

In-depth interviews were conducted with 30 patients selected in order to have a mix of patients with different types of diagnosis, age, gender, and level of adherence. Fourteen patients who were selected for interviews had DM (five with DM-1 and nine with DM-2), three had HTN only and 13 had DM and HTN. They were mostly females (n = 20, 67%) and six (20%) were <15 years.

Thematic content analysis identified multiple factors influencing adherence to DM and HTN medications. Those were factors related to the burden of disease and its treatment, including contextual and societal factors.

## Factors related to the disease and its treatment

The most important factor as a barrier to adherence was the person's perceived burden of disease and type of medications they were taking. Insulin, requiring an injection, was repeatedly perceived by those who used it as something that caused fear, pain, and strain or was hard to use due to side effects. However, those patients who were using pen insulin seemed to be positive about its use compared to syringes, expressing that it is easier to use and hence made them feel more comfortable. When a patient treatment was shifted from oral medication to insulin injection, they would expect improvement in their clinical outcomes.

> "The doctor told me he wants to give me insulin. I got scared. . . because one had to always inject himself with needles."

> Woman, 54 years, HTN and DM-2

> "He began to move more and eat, his tests were better. He felt normal. . . He used to reject the old insulin. I used to try a million way to convince him to take it."

> Woman, mother of an 11-year-old boy, DM-1

The chronicity of a disease like DM, together with the need to use insulin, seemed to be a double burden for patients and their caregivers. Some parents mentioned hiding from their children that insulin is a life-long medication in an attempt to increase adherence. The increased number of pills that a patient with DM or HTN must take also emerged as a factor that affected adherence in this population. Some misconceptions around potential negative consequences related to regularly taking their pills was mentioned, such as the fear of being addicted to the pills, which might lead a patient to interrupt taking them.

> "If I feel it is too much, I put a pause on the medication. I interrupted it for 2–3 days. I can't take that many pills."
>
> Woman, 59 years, DM-2

> "[. . .] I also had chest pains and I was prescribed more medication. I had a lot to take, so I stopped the diabetes medication."
>
> Woman, 58 years, DM-2 and HTN

### Factors related to self-perception of disease outcome

The results showed that while for many patients, having a stable glucose level or improved clinical outcomes due to a strict adherence to the prescribed treatment plan can be a motivating factor, having stabilized the disease with no apparent symptoms or seeing no improvement in clinical outcomes, these may have been a motive for some patients to discontinue their medications. Therefore, knowledge about the disease itself and its management would be an important factor affecting adherence. In relation to this, providing patients with PSEC was overall positively perceived and acknowledged as a factor in increasing patients' knowledge of their disease and its self-management including diet, as well as their understanding of their treatment plan.

> "I stopped taking my medication completely two months ago; my glucose level was good, so I neglected it."
>
> Woman, 58 years, DM-2

> "[I] do not see an improvement when I take the medication, but I keep on taking them because if I stop them my glucose and blood pressure will increase more."
>
> Woman, 50 years, DM-2 and HTN

> "[PSEC counselor] told me that if I take my medications regularly, my diabetes levels will decrease."
>
> Woman, 54 years, DM-2 and HTN

### Societal and contextual factors

Other factors affecting decreased adherence that emerged from the interviews were related to the patients' living context. Many had to work long hours in conditions that were not accommodating, making it challenging for those with even the best intentions to follow doctors' instructions. Also, often, patients needed to travel for clinic visits or work, finding themselves with interrupted treatment.

"When I go see my daughter, I take two injections with me. When they are finished, there will be nothing I can do. . . If I'm going somewhere far, I can't put it in the fridge."

Woman, 55 years, DM-2

For those who attend school, it can be challenging to maintain a high level of adherence if the school does not promote a culture of medical understanding. However, in our study, this did not seem to be an issue for most of the interviewees.

"[] The fact that she was the only one with diabetes at school did not affect her. If she needs to eat, she tells her teacher and leaves the class. They understand her situation there."

Woman, mother of an 8-year-old girl, DM-1

Family, peer and healthcare team support were important factors affecting self-adherence, and this was true for all ages. Specifically, for children, playing with friends was a priority and a motivating factor to take their medication. Good and trustworthy relationships between patients and the healthcare team were also important factors that affected adherence. Adherence was affected by the experience of other people who had the same disease or were taking the same medications. When patients identified themselves to others, this pushed them to question themselves or motivated them to be more adherent. The perception that family support of patients' treatment, specifically for those who were on insulin, was also a factor that affected the ability to keep up with their treatment plan.

"She saw how other kids who committed to their treatment felt better and how the un-committed felt worse. It pushed her to start working on herself."

Woman, mother of an 11-year-old girl, DM-1

"Once my son saw me [using insulin] and he started crying. He told me that I looked like I was taking drugs. If I was to die, I will not take insulin. . .my kids did not accept it."

Woman, 53 years, DM-2

"[I] feel she is more dependent on me. I believe she does remember when she should take her medication but if I do not remind her myself, she would ignore it."

Woman, mother of an 11-year-old girl, DM-1

## Discussion

To our knowledge, this study is the first to report the prevalence, risk factors and related clinical outcomes of adherence to DM and HTN medications in a Syrian refugee population with a high NCD burden. It is the only study in this protracted crisis context combining both quantitative and qualitative data and hence brings a more comprehensive understanding of the challenging and complex issue of medication adherence in this "real world" setting. It is also the first study in a refugee setting able to separately look and identify factors specific to adherence in type-1 DM patients in addition to those with type-2 DM.

The key findings in this study were: 1) the majority of patients were moderately or highly adherent to medications (almost 70%); 2) several factors positively affected adherence including peer and health professional support, PSEC, doctors' education and support, and insulin

pens provided to DM 1 patients ≤ 15 years old; 3) factors that negatively affected adherence included working hours, misconceptions about DM, stress related to precarious living conditions and using insulin, and the burden of treatment in general; 4) patients with a lower HbA1C were 75% more likely to be moderately to highly adherent. We believe this is a reflection of the combined patient-centered approach of standardized primary care and the MSF individualized patient support model [3, 26, 27]. Most of the factors influencing adherence were modifiable that could be improved through further education and utilizing a patient-centered approach.

Previous studies have looked primarily at the burden of insulin and oral medications for patients attending school, work, and the burden on elderly patients [28–30]. The level of adherence to medications seen in our study population seemed to be equal or higher than what was observed in the Middle East. For instance, a few studies conducted in Palestine on patients with DM using the MMAS questionnaire have shown an adherence rate for DM medications between 56% and 73% [31]. A review of patients with DM in the Middle East and North Africa conducted in 2017 concluded that the average medication low adherence in this region was of 38%, compared to the 30% reported in our study [24, 32, 33]. The countries of the Gulf, high-income countries, appeared to be more prone to low adherence with a reported average of 40% [18, 34].

The major predictors of adherence for diabetic patients shown by other studies were age, being on insulin and time since diagnosis [34]. These differences from our findings could be partially due to methodological differences between studies, as well as different populations.

In other studies with similar contexts working hours as well as misconceptions about medications were among the factors that affected diabetic adherence [35–37]. In our population, insulin was considered as a burden mainly due to the fear of injection and its stigma related to a feeling of embarrassment from injecting in public. This did not seem to be an issue with insulin pens which were shown to improve adherence in insulin-dependent patients [3, 26, 27, 38].

These findings demonstrate that even in some of the most challenging contexts, a patient-centered, team-based approach can be successfully implemented. It also found that the highest rate of moderate-high medication adherence overall was among those patients with type 1 DM (85%, p = 0.027). Not surprisingly, those with type 1 and 2 DM who were moderately-highly adherent had significantly greater frequency of reaching their target HbA1C ≤ 8.0%. We believe this is a reflection of the combined efforts of the healthcare team (providers and PSEC). Diabetic education, such as in the PSEC, has been repeatedly shown to improve patient outcomes [3, 26, 27].

The most common contributor to low adherence was the intentional cutting back of medication by patients when feeling worse/better. Special attention should be given to those patients who tend to decide to intentionally stop or extend their medications because they feel worse when they take it or because they feel their symptoms are under control. Therefore, we encourage further education and support to be provided to this subset of patients by providers and PSEC, in addition to those who are not meeting their HbA1C goal, in order to improve medication adherence. Emphasizing the importance of continuing their medication through education is critical and with messages specifically targeting potential intentional interruption of treatment may be helpful.

Also, more than 63% of our sample was on more than four medications, and this only included medications related to the chronic diseases that are followed at MSF clinic. Adherence generally drops off with more medications taken and the increased burden of the number of medication on adherence as shown in the Middle East and North Africa, but in this study,

there did not seem to be an increase of low adherence among those with more medications [32]. This may have been due to the extra care and education provided by PSEC.

This study showed the positive impact of PSEC on patients from a qualitative perspective. Patients referred to PSEC are generally the most challenging and more likely to have low medication adherence before receiving individual education and support. Education about the disease and its management has been shown in this study as well as in other settings targeting patients with DM and HTN to be effective in increasing adherence and improving patient outcomes [3, 26, 27, 39, 40]. This appeared to be important in the Middle East where a review of more than 30 studies looking at how knowledge, cultural and lifestyle beliefs influence the management of type-2 DM identified a lack of health knowledge impacted negatively on both medication adherence and clinical outcomes [40]. Hence and besides insulin access, we believe that patient-centered individual education and support is likely the most important "piece of the puzzle" in improving quality healthcare and outcomes especially in this context. The positive impact of PSEC on adherence was not reflected in the quantitative analysis, likely because patients seen in PSEC were a selective sample of those who were challenging to manage for providers and frequently requiring insulin.

Our study results provide valuable insights that could help improve care delivery, patient satisfaction, and clinical outcomes, particularly among refugees with unique challenges. These findings demonstrate that even in one of the most challenging contexts, a patient-centered, team-based approach can be successfully implemented.

In contrast, those with HTN were the least likely to be adherent (37%). Despite the low level of moderate and high adherence among those with just HTN, their overall level of blood pressure control was relatively good (average 131/78) and did not vary significantly from those with low adherence. This might reflect the fact that patients on antihypertensive medications, which have a longer half-life than insulin, can tolerate a lower rate of adherence with fewer detrimental outcomes [41].

Considering the context of Shatila and the many stressors facing people living as refugees in harsh conditions a moderate level of adherence is considered adequate and acceptable [3]. However, high adherence should remain the care team's goal in order to achieve optimal outcomes of care and prevention of potential complications among those with DM and HTN.

It was clear in our qualitative analysis that insulin use creates a substantial burden to patients and their families. This is important because many have recommended starting insulin early to avoid later complications among those with type 2 DM [42–44]. We feel a key recommendation is that, in this and similar contexts, it would be significantly less stressful on patients/families if more focus is placed on improving oral medication adherence and lifestyle changes to improve HbA1C control, rather than complicating management by adding insulin.

## Strengths & limitations

This study has many strengths:1) It included a unique context of a Syrian refugee population; 2) It combined both quantitative and qualitative evaluations to gain a more comprehensive understanding of a challenging and complex issue: medication adherence within a refugee community; 3) It was performed in a primary care "real world" setting rather than in a private location or higher income country; 4) Medications, clinical consultations and laboratory testing were all provided free of charge, removing to a large degree the cost of care, which can be a significant contributing factor to low adherence in other studies [3].

While this study brings new evidence in relation to adherence to NCD and HTN medications in the Syrian refugee setting, it also has limitations. Some of the known predictors for adherence, such as psychological factors, were not accounted for in the quantitative analysis.

As well, some of the factors shown to impact the glycemic control and consequently correlate with adherence, such as diet and exercise, were not considered. However, we do believe that those factors would potentially not have had a major influence on the adherence. In fact, we believe our population has a similar baseline when it comes to exercise and diet due to their refugee status and contextual challenges preventing them to have the luxury of choosing a diet that is suitable to their disease, and to move freely due to lack of legal documentation for many [45]. In addition, the choice of the presentation of the results used, whereby patients were compared by type of diagnosis, could have masked certain factors influencing adherence had we explored, for instance, drivers of adherence for patients on insulin versus those who are on oral medications. Similarly, the absence of significant predictors to moderate-high adherence in the multivariate analysis could possibly be due to a lack of statistical power limiting the ability to detect weaker associations [20]. Also, the number of interviews with HTN patients was limited suggesting that a larger sample size may have yielded different results. This was an uncontrolled study design, with its inherent limitations including self-reporting on the questionnaire, potential social desirability bias, and uncertainly regarding the recall period. Although these potential biases cannot be confirmed as no validated tool was used to measure them, we believe that the fact that the interviewer was known to the participants helped them talking freely about their issues in relation to medication adherence; the qualitative results showed they easily voiced their concerns and identified barriers to adherence. Social desirability bias may have affected the results for those who were 15 years or younger as their caregivers answered for them for the MMAS and qualitative interview. Also, the fact that the interviews were all conducted by a female, we cannot exclude a potential limitation regarding gender issues. The study results were context-specific and specific to the care provided by MSF that might be different than the care provided elsewhere, hence could not be generalized to other more traditional settings.

## Conclusion

This study has demonstrated that a good level of adherence to NCD medications in a refugee population can be achieved despite challenging circumstances. This may have been due to a patient-centered approach and educational support provided by the PSEC and MSF's comprehensive model of care to NCDs. Other contexts may find this model useful in addressing medication adherence. Future studies identifying additional factors and means of addressing barriers to adherence specific to the refugee population are still needed in order to allow reaching optimal levels of adherence and design well-informed intervention programs.

## Supporting information

**S1 File. Quantitative component questionnaire including the MMAS-8—English and Arabic versions.**
(DOCX)

**S2 File. Qualitative interview guide for patients–English and Arabic versions.**
(DOCX)

## Acknowledgments

The authors would like to acknowledge the management of the NCD MSF clinic in the Shatila camp as well as the coordination team who supported this study. We also want to thank the MSF Operational Research Center team in Luxembourg who supported this study technically.

We also want to thank our NCD patients for consenting to be part of the study and for their trust in MSF services and teams.

## Author Contributions

**Conceptualization:** Mariam Mohamad, Krystel Moussally, Chantal Lakis, Sola Bahous, Carla Peruzzo, Anthony Reid, Jeffrey K. Edwards.

**Data curation:** Mariam Mohamad, Krystel Moussally, Chantal Lakis.

**Formal analysis:** Mariam Mohamad, Krystel Moussally, Chantal Lakis.

**Investigation:** Mariam Mohamad.

**Methodology:** Mariam Mohamad, Krystel Moussally, Chantal Lakis, Sola Bahous, Anthony Reid, Jeffrey K. Edwards.

**Project administration:** Mariam Mohamad, Carla Peruzzo.

**Resources:** Mariam Mohamad, Krystel Moussally, Chantal Lakis, Anthony Reid, Jeffrey K. Edwards.

**Software:** Krystel Moussally, Chantal Lakis.

**Supervision:** Mariam Mohamad, Krystel Moussally, Chantal Lakis, Jeffrey K. Edwards.

**Validation:** Mariam Mohamad, Krystel Moussally, Chantal Lakis, Maya El-Hajj, Jeffrey K. Edwards.

**Visualization:** Mariam Mohamad, Krystel Moussally, Chantal Lakis, Maya El-Hajj, Sola Bahous.

**Writing – original draft:** Mariam Mohamad, Krystel Moussally, Chantal Lakis.

**Writing – review & editing:** Mariam Mohamad, Krystel Moussally, Chantal Lakis, Maya El-Hajj, Sola Bahous, Carla Peruzzo, Anthony Reid, Jeffrey K. Edwards.

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
