## [Decision Letter · Decision Letter 0]

22 Jan 2021

PONE-D-20-36223

Self-reported medication adherence among patients with diabetes or hypertension, Médecins Sans Frontières, Shatila Refugee Camp, Beirut, Lebanon: a mixed-methods study

PLOS ONE

Dear Dr. Mohamad,

Thank you for submitting your manuscript to PLOS ONE. After careful consideration, we feel that it has merit but does not fully meet PLOS ONE’s publication criteria as it currently stands. Therefore, we invite you to submit a revised version of the manuscript that addresses the points raised during the review process.

I received excellent, extensive feedback from two reviewers. Overall, the reviewing team thought that this mixed-methods examination of adherence in a refugee camp setting stands to make an important contribution to the literature. They offered several points for clarification and further discussion that I would ask you to please consider in a revision.

We look forward to receiving your revised manuscript.

Kind regards,

Rachel A. Annunziato, Ph.D.

Academic Editor

PLOS ONE

Journal Requirements:

2. Thank you for your submission to PLOS ONE. PLOS requires that a “minimal data set” is shared, defined as the data set used to reach the conclusions drawn in the manuscript with related metadata and methods, and any additional data required to replicate the reported study findings in their entirety. Authors do not need to submit their entire data set if only a portion of the data were used in the reported study. Also, authors do not need to submit the raw data collected during an investigation if the standard in the field is to share data that have been processed. Please submit the following data: The values behind the means, standard deviations and other measures reported; The values used to build graphs; The points extracted from images for analysis.” Please review http://journals.plos.org/plosone/s/data-availability#loc-faqs-for-data-policy.

3. Please include additional information regarding the survey and qualitative questionnaires used in the study and ensure that you have provided sufficient details that others could replicate the analyses. For instance, if you developed a questionnaire as part of this study and it is not under a copyright more restrictive than CC-BY, please include a copy, in both the original language and English, as Supporting Information, or include a citation if it has been published previously.

"The authors would like to acknowledge the management of the NCD MSF clinic in the Shatila

 camp as well as the coordination team who supported this study. We also want to thank the MSF

Operational Research Center team in Luxembourg who were behind this study."

"The author(s) received no specific funding for this work"

6. Please note that in order to use the direct billing option the corresponding author must be affiliated with the chosen institute. Please either amend your manuscript to change the affiliation or corresponding author, or email us at plosone@plos.org with a request to remove this option.

7. PLOS requires an ORCID iD for the corresponding author in Editorial Manager on papers submitted after December 6th, 2016. Please ensure that you have an ORCID iD and that it is validated in Editorial Manager. To do this, go to ‘Update my Information’ (in the upper left-hand corner of the main menu), and click on the Fetch/Validate link next to the ORCID field. This will take you to the ORCID site and allow you to create a new iD or authenticate a pre-existing iD in Editorial Manager. Please see the following video for instructions on linking an ORCID iD to your Editorial Manager account: https://www.youtube.com/watch?v=_xcclfuvtxQ

8. We note that you have included the phrase “data not shown” in your manuscript. Unfortunately, this does not meet our data sharing requirements. PLOS does not permit references to inaccessible data. We require that authors provide all relevant data within the paper, Supporting Information files, or in an acceptable, public repository. Please add a citation to support this phrase or upload the data that corresponds with these findings to a stable repository (such as Figshare or Dryad) and provide and URLs, DOIs, or accession numbers that may be used to access these data. Or, if the data are not a core part of the research being presented in your study, we ask that you remove the phrase that refers to these data.

9. Please include a copy of Table 3 which you refer to in your text on page 16.

Reviewers' comments:

Reviewer's Responses to Questions

**Comments to the Author**

1. Is the manuscript technically sound, and do the data support the conclusions?

Reviewer #1: Yes

Reviewer #2: Yes

2. Has the statistical analysis been performed appropriately and rigorously? 

Reviewer #1: Yes

Reviewer #2: Yes

3. Have the authors made all data underlying the findings in their manuscript fully available?

Reviewer #1: No

Reviewer #2: No

4. Is the manuscript presented in an intelligible fashion and written in standard English?

Reviewer #1: Yes

Reviewer #2: Yes

5. Review Comments to the Author

Reviewer #1: Thank you for the opportunity to review this interesting manuscript.

1. At the outset, the authors need to report, if there exist any overlap of results with reference 3 (Kayali et al)

2. Methods:

Need further site description-

What is the duration of the medication refill provided at the MHF facility, and what is the frequency of prescribed follow-up recommended for the patients? If 43 patients did not report for their scheduled follow-up, did the patients stop visiting the facility for any reasons, since otherwise you could have collected the data from them on their next visit. Furthermore, do the patients acquire their DM medications from any other site apart from the MHF clinic?

There are several methods of medication adherence including objective measures such as pill counts, and database analysis (See, Basu et al: 2019; https://www.ncbi.nlm.nih.gov/pmc/articles/PMC6450154/). Since, these methods were not used in this study, it becomes important to identify the adequacy of drug stocks with the patients.

The MMAS-8 is a self-administered instrument, but considering the low-literacy in your sample, it should have been verbally administered. You should cite a reference in support of the validity of verbal administration of the scale among the illiterate or low-literacy patients (Example: Basu et al. https://doi.org/10.1007/s13410-014-0232-9)

3. Results:

The age-categories (<15) are arbitrary. Usually, age-categories are differentiated based on median values or on other physiological criterion

Polypharmacy and increased regimen complexity are usually associated with poor medication adherence (Odegard et al: 2007; doi: 10.1177/0145721707308407). In this context, did you find any variations in adherence among multimorbid patients.

Who paid for the insulin pens? If they were provided with MHF, then why was their use not universal among insulin using patients?

Did you assess knowledge of diabetes in the patients using any standardized instrument, since it was an important predictor of glycemic control?

Diet and exercise adherence would also influence glycemic control and occasionally may correlate with medication adherence indicative of a higher self-efficacy (Basu et al: 2015). Why were these parameters not assessed in this study using a suitable study instrument like the SDSCA - mention this is a limitation.

<patients 75="" a="" hba1c="" lower="" were="" with="">

I would suggest to reframe this sentence and report the odds instead

Family support can improve medication adherence especially in the elderly and disabled such as those with low vision. Did you encounter such a finding in your qualitative analysis?

Discussion section should be shortened. Avoid suggesting an association without data such as the possibility of increased self-desirability bias among adolescents. Questionnaire tools to measure self-desirability in populations do exist.

You have reported lower adherence to Hypertensive medication among the ?comorbid patients - compare and contrast with the global evidence.

Conclusion needs to be rephrased. For future studies, what do you recommended?</patients>

Reviewer #2: This is an important and well executed study that should be published. The work contributes to a sparse evidence base on an important and timely topic. The article is well written with mostly minor suggestions for improvement. See detailed comments in the uploaded review attachment.

6. PLOS authors have the option to publish the peer review history of their article (what does this mean?). If published, this will include your full peer review and any attached files.

Reviewer #1: **Yes: **Saurav Basu

Reviewer #2: No

---

## [Author Response · Author response to Decision Letter 0]

1 Apr 2021

Dear Editor, 

Thank you for giving me the opportunity to submit a revised draft of my manuscript titled Self-reported medication adherence among patients with diabetes or hypertension, Médecins Sans Frontières Shatila Refugee Camp, Beirut, Lebanon: a mixed-methods study to Plos one journal. We appreciate the time and effort that you and the reviewers have dedicated to providing your valuable feedback on our manuscript. We are grateful to the reviewers for their insightful comments on our paper. We have been able to incorporate changes to reflect most of the suggestions provided by the reviewers. Also, we have highlighted the changes within the manuscript. 

Here is a point-by-point response to the reviewers’ comments and concerns.

Point-by-point response to the reviewers’ reports:

Reviewer #1: 

1.At the outset, the authors need to report, if there exist any overlap of results with reference 3 (Kayali et al)

Response:

Thank you for raising this point. Both this study and reference #3 in the manuscript (Kayali et al.), were conducted in the same MSF NCD clinic following the same model of care and had the patients included selected from the same population source; however, they had different aims, objectives and methodology, and hence have yielded different results. This study focused on measuring the self-adherence in patients with diabetes and hypertension and assessed the factors affecting their adherence, while the study of Kayali et al. aimed at looking at the outcome of care to patients with diabetes and hypertension followed at the MSF NCD clinic in Shatila. What might have been similar to both studies is the characteristics of the study population as both were selected randomly. However, the timeline of the two studies was different, therefore contextual factors might have affected a possible change in the population accessing care in the MSF clinic in Shatila.

We added a specification at line 77-78 of the track changed version of the article and as suggested by the reviewer, the corresponding sentence now reads: “While a recent study, different in its scope and results, conducted at the MSF Shatila clinic showed promising outcomes of care in DM and HTN patients, it has been challenging for MSF providers to achieve optimum control in these patients.”

2. Methods:

2.1. Need further site description-

What is the duration of the medication refill provided at the MHF facility, and what is the frequency of prescribed follow-up recommended for the patients? If 43 patients did not report for their scheduled follow-up, did the patients stop visiting the facility for any reasons, since otherwise you could have collected the data from them on their next visit. Furthermore, do the patients acquire their DM medications from any other site apart from the MHF clinic?

Response:

We thank the reviewer for pointing this out. In an attempt to give more details in terms of the site description as suggested by the reviewer, we added the following under the sub-section “MSF model of care” in the Methods section: 

Lines 165-169 of the track changed version of the article:

“Patients followed at our center and who are stable as per their blood pressure -for patients with HTN- and/or their blood sugar readings -for those with DM-, have their medical consultation and medication refill taking place every 2 to 3 months. Those who are unstable, are requested to present for an appointment within a month in order to closely follow them up and monitor their outcomes and treatment plans.” 

We invited the readers to read further details and specifications of the model of care in the Kayali et al., article as referenced in this section.

For the 43 patients who did not show up at the time of data collection, we did not check if they actually re-visited the clinic at a later stage. However, knowing that a good proportion of our patients do not show up at the exact date of the scheduled appointment, we did take this into consideration in the sample size calculation as stated in line 203-204 of the revised track changed version of the article regarding the original sample being inflated by: “and an additional 10% to account for no-shows at the time of the scheduled appointment” 

Regarding the possibility that DM patients might have been acquiring their DM medications from other sites, it is not possible to tell. However, we believe that this is unlikely as our patients receive all their needed medications from the MSF center free of charge. They can, however, be visiting specialists for potential other comorbidities such as cardiovascular diseases not being followed by MSF teams and hence might need to buy additional medications not available in MSF sites.

2.2. There are several methods of medication adherence including objective measures such as pill counts, and database analysis (See, Basu et al: 2019; https://www.ncbi.nlm.nih.gov/pmc/articles/PMC6450154/). Since, these methods were not used in this study, it becomes important to identify the adequacy of drug stocks with the patients. The MMAS-8 is a self-administered instrument, but considering the low-literacy in your sample, it should have been verbally administered. You should cite a reference in support of the validity of verbal administration of the scale among the illiterate or low-literacy patients (Example: Basu et al. https://doi.org/10.1007/s13410-014-0232-9)

Response: We do agree with the reviewer that there are several methods to assess medication adherence. The MMAS questionnaire was chosen in this study as it had previously been used in the region as referenced through the article, and in contexts that are similar to this one. This allowed comparability of results. It would have been ideal, as suggested by the reviewer, to validate medication adherence with another measure such as the pill counts, but this was not accounted for in the study methodology. 

As suggested by the reviewer, we have added two references supporting the verbal administration of the of the MMAS questionnaire in communities with literacy challenges. These are in lines 216-219 of the revised track changed version of the article, reading: “For patients and caregivers who were unable to read and/or write, the MMAS-8 questionnaire was verbally administered. The verbal administration of the MMAS-8 questionnaire has been previously used in other studies in communities with lack of or low-literacy level.”

3. Results:

3.1. The age-categories (<15) are arbitrary. Usually, age-categories are differentiated based on median values or on other physiological criterion

Response:

It is true that the age categories are usually differentiated based on the median values of other physiological criteria. In MSF projects the cut-off of 15 is used to differentiate pediatric from non-pediatric communities. This cut-off is used in our monitoring data and indicators. Therefore, to allow an MSF operational interpretation of the results, we left the age cut-off at 15 years old. In addition, we wanted to assess differences in adherence between children (defined as being <=15 years old) and adults (defined as >15 years old). Also, patients who are ≤ 15 years old are the only ones who use insulin pens while the others use either oral medications or another type of Insulin; therefore age is a proxy indicator of the form of insulin used. 

3.2. Polypharmacy and increased regimen complexity are usually associated with poor medication adherence (Odegard et al: 2007; doi: 10.1177/0145721707308407). In this context, did you find any variations in adherence among multimorbid patients.

Response

As presented as part of the bivariate analysis in table 3 labelled “Table 3. Bivariate analyses among low adherence (MMAS-8 score <6) and moderately/high adherence (MMAS-8 score≥6) for study participants, Shatila primary care clinic, Beirut, October 2018”, there was no statistically significant association between the level of adherence and the number of chronic co-morbidities. However, results showed -as stated in the results section of the track changed version of the revised article at lines 325-326of the “Bivariate analysis section- and as showed in Table 3, “Being on two chronic medications seemed to be significantly increasing the likelihood of being moderately to highly adherent.” This was also identified in the qualitative results as stated in lines 357-359 of the tracked changed version of the revised article: “The increased number of pills that a patient with DM or HTN must take also emerged as a factor that affected adherence in this population.” 

This was also the case for the number of chronic medications prescribed. However, since both variables are specific to the chronic morbidities and chronic medications of interest (followed-up at MSF clinic), they might not reflect the total number of morbidities and chronic medications the patients are on as that data is not systematically recorded in the electronic system used in this study. This was mentioned in the Discussion section of the article in Lines 482-483 of the track changed version revised article stating that “…this only included medications related to the chronic diseases that are followed at MSF clinic.”

3.3. Who paid for the insulin pens? If they were provided with MHF, then why was their use not universal among insulin using patients?

Response

All medications mentioned in the study were provided free of charge by MSF including Insulin pens. However, due to their expense, insulin pens were provided only to patients with DM type 1 who were under 15 years old (MSF definition of children). 3.4. Did you assess knowledge of diabetes in the patients using any standardized instrument, since it was an important predictor of glycemic control?

Response:

In this study we did not assess patients’ knowledge of their treatment and disease. Although this is important as a baseline for the Patient Support and Education Counselling (PSEC), a variable considered in this study, we believe that it is not under the scope of this study. In usual practice, and when a patient is referred for PSEC, their baseline knowledge is assessed through a qualitative interview, the results of which are used to identify what and where to focus on during the PSEC sessions. 

3.5. Diet and exercise adherence would also influence glycemic control and occasionally may correlate with medication adherence indicative of a higher self-efficacy (Basu et al: 2015). Why were these parameters not assessed in this study using a suitable study instrument like the SDSCA - mention this is a limitation.

Response:

Thank you for mentioning the use of an assessment tool for diet and exercise such as the SDSCA. This was not planned as part of the study design. However, we do not believe that would have impacted the results, but to our knowledge, it does not directly influence adherence. In addition, we do believe that our population had a similar baseline when it came to diet and exercise. Patients would be comparable since 1) they could not really choose a diet that was not limited by finances and availability, and 2) they were not able to exercise as they lived in a very crowded environment with security challenges (Reference: United Nations High Commission for Refugees (UNHCR), United Nations population and Children's fund (UNFPA), World food Programme (WFP). Vulnerability assessment of Syrian refugees in Lebanon, VASYR 2017. 2017).

As suggested by the reviewer, we did add this to the limitations of the study. Lines 542-548 in the “Strengths and Limitations” sub-section of the discussion in the track changed version of the manuscript reads: “As well, some of the factors shown to impact the glycemic control and consequently correlate with adherence, such as diet and exercise, were not considered. However, we do believe that those factors would potentially not have had a major influence on the adherence. In fact, we believe our population has a similar baseline when it comes to exercise and diet due to their refugee status and contextual challenges preventing them to have the luxury of choosing a diet that is suitable to their disease, and to move freely due to lack of legal documentation for many.”

3.6. I would suggest to reframe this sentence and report the odds instead

It is not clear which sentence is being reference (no line number provided). Please provide more details.

3.7. Family support can improve medication adherence especially in the elderly and disabled such as those with low vision. Did you encounter such a finding in your qualitative analysis?

Response: 

Thanks for pointing out this. Based on our qualitative results and what we have heard in the interviews, family support was mostly offered to children less than 15 years who were on insulin. No specific findings in terms of family support for elderly or disabled was identified. 

4. Discussion 

4.1. Discussion section should be shortened. Avoid suggesting an association without data such as the possibility of increased self-desirability bias among adolescents. Questionnaire tools to measure self-desirability in populations do exist.

Response

We do agree with the reviewer that social desirability bias was a potential bias in the study and so we added the following sentence: “Although these potential biases cannot be confirmed as no validated tool was used to measure them, we believe…” in lines 557-558 in the strengths and limitations sub-section of the Discussion of the track changed version of the article.

As suggested by the reviewer, we did remove some sentences from the Discussion to shorten it as shown in the track change version of the article.

4.2. You have reported lower adherence to Hypertensive medication among the comorbid patients - compare and contrast with the global evidence.

Response:

As reported in table 3 of the manuscript, 37% of patients with HTN are low adherent compared to 35% for those with DM and HTN. While the difference among the 4 categories of co-morbidities (DM-1 only, DM-2 only, HTN only, and DM+HTN) in relation to adherence was statistically significant, the one between the two categories of HTN alone and DM+HTN was relatively small. Therefore, it was not discussed in comparison to global evidence to emphasize on other results, deemed more important. . 

5. Conclusion 

Conclusion needs to be rephrased. For future studies, what do you recommended?

Response

As suggested by the reviewer, the conclusion has been rephrased and recommendations for future studies have been included. The conclusion in the track changed version of the manuscript, lines 586-592, now reads: “This study has demonstrated that a good level of adherence to NCD medications in a refugee population can be achieved despite challenging circumstances. This may have been due to a patient-centered approach and educational support provided by the PSEC and MSF’s comprehensive model of care to NCDs. Other contexts may find this model useful in addressing medication adherence. Future studies identifying additional factors and means of addressing barriers to adherence specific to the refugee population are still needed in order to allow reaching optimal levels of adherence and design well-informed intervention programs.

Reviewer # 2:

1.Introduction 

1.1. Introduction (line 84): Clarify that the 50% low adherence estimate is global not population-specific

Response: 

Thank you for pointing this out. Lines 81-82 of the track changed version of the article now reads: “One of the determinants of poor NCD outcomes is low adherence to medication, which has been estimated globally to be 50% in patients with DM and/or HTN, making it a primary risk factor for poor outcomes in these diseases”.

1.2. Introduction (lines 93-94): There is also a relatively new (2020) article on exactly this: https://doi.org/10.1007/s40200-020-00638-6

Response:

Many thanks for indicating the new reference. We adjusted our sentence to include the article proposed. The following was added to the introduction section of the track changed article in lines 96-98: “Another study recently published mentioned that around 25% of Syrian refugees in Lebanon had to interrupt their HTN or DM medication in the year preceding the survey due to costs.”

2.Methods 

2.1. Methods (line 130): For those on insulin and oral medication, is “adherence” to one or both?

Response: 

For those patients on insulin and oral medications, there was no distinction regarding adherence based on the MMAS questionnaire. Therefore it represents adherence to the DM medications patients were on. 

2.2. Methods (“MSF model of care” & generally throughout): To what extent do MSF patients receive care or medication outside MSF clinic(s)? This may not be known but is a question that came up for me throughout, so may be worth addressing, if possible, along with implications.

Response: 

MSF treats patients with diabetes, asthma, cardiovascular diseases, epilepsy, and hypothyroidism. Since care provided is free of charge (medications, consultations, laboratory) for these diseases, and the population cared for is vulnerable, we do not expect that our patients seek care elsewhere. They might seek specialized care that is not provided in our center or in case of presence of a comorbidity that is not part of the list of diseases that MSF is capable of covering. This specification was added in lines 157-159 if the track changed article: “Since care provided in MSF center is free of charge for the above-mentioned diseases, we believe that the patients followed-up for care at our center do not seek care elsewhere. However, this cannot be verified.”

2.3. Methods (line 217): Suggest mentioning in the limitations section potential limitation regarding gender issues since a female conducted all interviews and possible gratuity bias since research was done by known providers/at clinic.

Response:

Thank you for pointing this out. However we believe that “the fact that the interviewer was known to the participants helped them talking freely about their issues in relation to medication adherence” as per lines 560-561 of the track changed version. As well, even though the fact that a female conducted all interviews, and there might be gender issues as per lines 562-563 of the track changed version, the qualitative results showed that the patients and caregivers easily voiced their concerns and identified barriers to adherence. However, since this cannot be excluded, we added this as a limitation, lines 564-565 of the track changed version. “Also, the fact that the interviews were all conducted by a female, we cannot exclude a potential limitation regarding gender issues.”

2.4. General comment on analysis and presentation of results: It is complicated to lump insulin with oral Rx adherence; the issues and drivers of adherence are often different. I would suggest exploring analysis separately for insulin vs. oral and including as supplementary table(s). At a minimum, the potential influence of analyzing these together should be mentioned. 

Response:

We do agree with the reviewer that the drivers of adherence are different between patients who are on insulin versus those who are in oral medications. This result was actually highlighted in the qualitative analysis and was implied in the quantitative analysis that showed patients with type-1 DM were most likely to be moderately to highly adherent compared to other disease categories. We believe that using the dichotomous variable “on insulin” in the descriptive analysis as well as the bivariate and regression analysis gives a good reflection on the specificity of insulin use compared to oral medications. It would have definitely been interesting, as suggested by the reviewer, to explore the results separately for insulin versus oral medications, independent of the disease; however, we believed that the choice of going for this way of presenting the results is the best choice translating into potential recommendations to improve adherence in our patients. 

As suggested by the reviewer, we did add in the Discussion lines 549-551 of the track changed version: “In addition, the choice of the presentation of the results used, whereby patients were compared by type of diagnosis, could have masked certain factors influencing adherence had we explored, for instance, drivers of adherence for patients on insulin versus those who are on oral medications.”

3.Results 

3.1. Results (line 308): How many of those on >1 medication were on insulin + oral?

Response:

Unfortunately, it is not possible to identify the total number/proportion of patients who are on insulin and oral medications. The data collection was not carried out in a way to allow pulling out of this number.

3.2. Results (“Multivariate Analysis” section): The absence of significant predictors may also have been related to distribution of predictors/adherence since the sample size is not so large, limiting the ability to detect weaker associations.

Response:

We agree with the reviewer. We added this as a potential limitation in the “Strengths and limitations” sub-section of the methods in lines 552-554 of the tracked version of the article, reading: “Similarly, the absence of significant predictors to moderate-high adherence in the multivariate analysis could possibly be due to a lack of statistical power limiting the ability to detect weaker associations.”

3.3. Results (lines 352-3): It is unclear if this is from the current study findings or previous studies/assumptions

Response:

We realize this sentence is confusing. We changed it to read, in lines 375-379 of the tracked version of the article: “The results showed that while for many patients, having a stable glucose level or improved clinical outcomes due to a strict adherence to the prescribed treatment plan can be a motivating factor, having stabilized the disease with no apparent symptoms or seeing no improvement in clinical outcomes, these may have been a motive for some patients to discontinue their medications.”

4. Discussion 

4.1. Discussion (lines 499-502): Wording in this sentence is confusing

Response:

Thanks for this comment. We changed the wording of the sentence to make it clear. It now reads, in lines 519-521 of the track changed version: “This might reflect the fact that patients on antihypertensive medications, which have a longer half-life than insulin, can tolerate a lower rate of adherence with fewer detrimental outcomes.”

4.2. Strengths & Limitations (lines 518-20): True but removing one of the well-known drivers of low adherence (i.e., cost) may also be a limitation

Response:

We agree that removing a well-known driver for low-adherence being cost of treatment and follow-up may be a limitation in a sense that the proportion of low adherence might be expected to be higher in settings where patients have to pay out-of-pocket money. However, we believe that in a refugee setting, if costs were not covered, they would mask other factors and presented as a leading cause for low-adherence. 

4.3. Suggest mentioning limiting the sample to MSF patients as a limitation in terms of representativeness; those not receiving care from MSF may have different adherence prevalence and predictors

Response: 

Based on the reviewer suggestion, we modified the last sentence in the discussion, lines 569-571 of the track changed version to become: “The study results were context-specific and specific to the care provided by MSF that might be different than the care provided elsewhere, hence could not be generalized to other more traditional settings.”

5. Conclusion 

Conclusion (lines 537-8): Does this sentence draw directly from the study findings? This claim’s basis would require external reference(s), no? 

Response:

We agree the first sentence of the conclusion is confusing and does not directly draw from the findings. Therefore, it was removed and the conclusion re-worded based on the second reviewer’s comment.

---

## [Decision Letter · Decision Letter 1]

26 Apr 2021

Self-reported medication adherence among patients with diabetes or hypertension, Médecins Sans Frontières, Shatila Refugee Camp, Beirut, Lebanon: a mixed-methods study

PONE-D-20-36223R1

Dear Dr. Mohamad,

We’re pleased to inform you that your manuscript has been judged scientifically suitable for publication and will be formally accepted for publication once it meets all outstanding technical requirements.

Kind regards,

Rachel A. Annunziato, Ph.D.

Academic Editor

PLOS ONE

Additional Editor Comments (optional):

Reviewers' comments:

Reviewer's Responses to Questions

**Comments to the Author**

1. If the authors have adequately addressed your comments raised in a previous round of review and you feel that this manuscript is now acceptable for publication, you may indicate that here to bypass the “Comments to the Author” section, enter your conflict of interest statement in the “Confidential to Editor” section, and submit your "Accept" recommendation.

Reviewer #1: All comments have been addressed

Reviewer #2: All comments have been addressed

2. Is the manuscript technically sound, and do the data support the conclusions?

Reviewer #1: Yes

Reviewer #2: Yes

3. Has the statistical analysis been performed appropriately and rigorously? 

Reviewer #1: Yes

Reviewer #2: Yes

4. Have the authors made all data underlying the findings in their manuscript fully available?

Reviewer #1: Yes

Reviewer #2: Yes

5. Is the manuscript presented in an intelligible fashion and written in standard English?

Reviewer #1: Yes

Reviewer #2: Yes

6. Review Comments to the Author

Reviewer #1: All comments have been reasonably addressed by the authors. However, i recommended some language editing for further improvement.

Reviewer #2: (No Response)

7. PLOS authors have the option to publish the peer review history of their article (what does this mean?). If published, this will include your full peer review and any attached files.

Reviewer #1: **Yes: **Saurav Basu

Reviewer #2: No

---

## [Editor Report · Acceptance letter]

29 Apr 2021

PONE-D-20-36223R1 

Self-reported medication adherence among patients with diabetes or hypertension, Médecins Sans Frontières Shatila Refugee Camp, Beirut, Lebanon: a mixed-methods study 

Dear Dr. Mohamad:

I'm pleased to inform you that your manuscript has been deemed suitable for publication in PLOS ONE. Congratulations! Your manuscript is now with our production department. 

Kind regards, 

on behalf of

Dr. Rachel A. Annunziato 

Academic Editor

PLOS ONE